# HIIT vs. SIT: What Is the Better to Improve $\dot{V}$O_2_max? A Systematic Review and Meta-Analysis

**DOI:** 10.3390/ijerph182413120

**Published:** 2021-12-12

**Authors:** Silas Gabriel de Oliveira-Nunes, Alex Castro, Amanda Veiga Sardeli, Claudia Regina Cavaglieri, Mara Patricia Traina Chacon-Mikahil

**Affiliations:** 1Exercise Physiology Laboratory, University of Campinas (UNICAMP), Av. Érico Verissimo, 701-Cidade Universitária “Zeferino Vaz” Barão Geraldo, Campinas 13083-851, SP, Brazil; cavaglieri@fef.unicamp.br (C.R.C.); marapatricia@fef.unicamp.br (M.P.T.C.-M.); 2Nuclear Magnetic Resonance Laboratory, Federal University of São Carlos (UFSCar), Rod. Washington Luiz, s/n, São Carlos 13565-905, SP, Brazil

**Keywords:** HIIT, SIT, $\dot{V}$O_2_max, interval training, HIIT vs. SIT

## Abstract

Lack of time is seen as a barrier to maintaining a physically active lifestyle. In this sense, interval training has been suggested as a time-efficient strategy for improving health, mainly due to its potential to increase cardiorespiratory fitness. Currently, the most discussed interval training protocols in the literature are the high-intensity interval training (HIIT) and the sprint interval training (SIT). Objective: We investigated, through a systematic review and meta-analysis, which interval training protocol, HIIT or SIT, promotes greater gain in cardiorespiratory fitness ($\dot{V}$O_2_max/peak). The studies were selected from the PubMed (MEDLINE), Scopus and Web of Science databases. From these searches, a screening was carried out, selecting studies that compared the effects of HIIT and SIT protocols on $\dot{V}$O_2_max/peak. A total of 19 studies were included in the final analysis. Due to the homogeneity between studies (I^2^ = 0%), fixed-effects analyses were performed. There was no significant difference in the $\dot{V}$O_2_max/peak gains between HIIT and SIT for the standardized mean difference (SMD = 0.150; 95% CI = −0.038 to 0.338; *p* = 0.119), including studies that presented both measurements in mL·kg^−1^·min^−1^ and l·min^−1^; and raw mean differences (RMD = 0.921 mL·kg^−1^·min^−1^; 95% CI = −0.185 to 2.028; *p* = 0.103) were calculated only with data presented in mL·kg^−1^·min^−1^. We conclude that the literature generates very consistent data to confirm that HIIT and SIT protocols promote similar gains in cardiorespiratory fitness. Thus, for this purpose, the choice of the protocol can be made for convenience.

## 1. Introduction

Lack of time has been acknowledged as one of the main barriers to maintaining a physically active lifestyle in humans [1]. Physical activity promotes and maintains health by improvement of cardiorespiratory fitness, for which the levels are inversely related to mortality from different causes [2]. International guidelines recommend the practice of at least 150 min per week for continuous moderate cardiorespiratory training or 75 min per week of vigorous intensity to promote health benefits [3].

In this sense, high-intensity interval training (HIIT) has been proposed as a time-efficient alternative strategy for maintaining a physically active lifestyle, since it promotes benefits equal to or even greater than traditional continuous aerobic training [4,5,6]. HIIT consists of performing repeated series of high-intensity efforts (usually between the second ventilatory threshold and the $\dot{V}$O_2_max/peak) interspersed with periods of low-intensity recovery/or pause [1,7,8,9]. In this perspective, variations of HIIT protocols have emerged, applying intensities and/or loads above the $\dot{V}$O_2_max/peak in very short periods of time. Here these short-bout protocols are referred to as sprint interval training (SIT) [1,10].

The prescription of either HIIT or SIT is complex and involves the manipulation of several variables, including duration of the effort and recovery phases, intensity of effort and recovery, and total number of efforts/recovery [9,10,11].

From the manipulation of these variables, a range of protocols may be designed, based on well-known protocols, such as Wingate, which consists of performing efforts lasting 30 s at all-out intensity, to protocols using intensities close to maximum (according to different forms of exercise prescription, e.g., $\dot{V}$O_2_max, maximum power output, Vmax, etc.) [10] and duration around to 4 min [4,12]. The differences in the training variables between HIIT and SIT include time of effort and intensity, which are expected to reflect changes in the metabolism and adaptations of organic systems [11]. When comparing these protocols, Matsuo et al. (2014) demonstrated a significant increase in $\dot{V}$O_2_max after 8 weeks of HIIT and SIT, evidencing that both programs are effective to improve cardiorespiratory fitness [13].

Previous studies seem to confirm that the manipulation of duration and intensity of the effort determines the energetic and metabolic stress of the exercise session, which constitute the driving force that triggers physiological and cellular processes that, in turn, lead to chronic adaptation to exercise training [14,15].

Higher gains in the $\dot{V}$O_2_max have been associated with training at longer-duration bouts close to that of the $\dot{V}$O_2_max [10,11,16]. This phenomenon is dependent on the increase of the exercise intensity, duration of the stimulus, recruitment/activation of types I and II muscle fibers and the use of energetic substrates [17]. Thus, different intensities and duration of effort have different impacts on the organic systems. For example, while bouts at maximum/supramaximal intensities with short duration should also affect the neuromuscular system, bouts at submaximal/near-to-maximum intensities mainly improve cardiovascular and cardiorespiratory fitness [10]. These findings led us to hypothesize that different durations in HIIT effort phase lead to different metabolic and organic system demands, thus leading to different adaptations in relation to $\dot{V}$O_2_max.

Therefore, the aim of this study was to investigate, through a systematic review and meta-analysis, which interval training protocol, HIIT or SIT, promotes greater gain in cardiorespiratory fitness ($\dot{V}$O_2_max/peak).

## 2. Materials and Methods

### 2.1. Search Strategy

Searches were conducted in October 2021, using articles from PubMed (MEDLINE), Scopus and Web of Science databases, with the following descriptors for protocols: high-intensity interval training, high-intensity interval exercise, high-intensity interval aerobic, interval training, sprint interval training, high-intensity intervals, repeat sprint training, high-intensity intermittent exercise, repeated sprint exercise, high-intensity intermittent exercise and aerobic interval training. The descriptors for duration of effort were exercise dose, short extent, long extent, long time, short time, long volume and short volume. The descriptors for the outcomes were maximal oxygen uptake, peak oxygen uptake, cardiorespiratory fitness and $\dot{V}$O_2_max/peak. For the combination of synonyms within the same category, “OR” was used between descriptors, while for combinations between the different groups of categories, “AND” was used.

The studies were screened by two research evaluators to select those containing both HIIT and SIT protocols. Quality assessment was performed by using the TESTEX scale (Table 1) that was also carried out by two evaluators [18]. To complement the number of studies included, a manual search was also carried out by consulting the list of references of the included studies. This study was reported by following the guidelines of the Preferred Reporting Items for Systematic and Meta-Analyses (PRISMA) [19].

### 2.2. Inclusion and Exclusion Criteria

#### 2.2.1. Type of Studies

Studies written in English that compared HIIT to SIT training were included in this meta-analysis. Articles not containing data from both the pre- and post-$\dot{V}$O_2_max or peak ($\dot{V}$O_2_max/peak), conducted in animals or review studies were excluded. Studies were also excluded when HIIT and SIT protocols were not directly compared and they presented their data in a graphical mode that did not allow for data extraction via software.

#### 2.2.2. Participants

No restrictions were applied to the different ages, sex, body composition, comorbidities and levels of physical activity of the subjects included in each survey. Studies involving subjects with spinal cord injuries or individuals with special health conditions were excluded.

#### 2.2.3. Protocols

Studies with interventions lasting at least two weeks, with a weekly frequency of at least two days, were selected. Studies that did not present measurements of $\dot{V}$O_2_max/peak before and after the intervention period were excluded. To be included, studies should have subjects assigned to both the HIIT group and the SIT group. In the present study, for the purpose of comparing protocols, we made use of the classification suggested by Bucchet and Laursen (2013) [10] that considers efforts lasting less than 60 s as short duration, here called SIT (sprint interval training), while longer efforts (> 60 s) are considered long duration and are here named HIIT (high-intensity interval training). Thus, HIIT refers to protocols performed with effort lasting over 60 s up to 5 min, in an intensity close to the maximum, maximum or above the maximum (> 80% of the maximum intensity), according to the different forms of evaluation and prescription intensity (PPO, Vmax, $\dot{V}$O_2_max, HR, etc.).

### 2.3. Data-Extraction Strategy

The main outcomes data of each time-point and group and characteristics of the selected population, such as age, sex, height, weight, physical activity level, health conditions and the $\dot{V}$O_2_max or peak, were manually extracted for descriptive and statistical analysis purposes. Data were recorded as mean and standard deviation. In studies that did not present such data, the authors were contacted to provide the information.

### 2.4. Quality Assessment and Publication Bias

Egger’s test, together with the visual analysis of the funnel plot, was used to analyze the risk of publication bias [20].

### 2.5. Choice of Model and Analysis of Heterogeneity

The choice of the analysis model was made by observing the heterogeneity between the studies. For this, the I-square (I^2^) test and Cochran’s Q test and *p*-values were taken as parameters for this decision. Different I^2^ values above 25% were adopted as an indicator of significant heterogeneity between studies; however, the *p*-value > 0.05 of the Q test was analyzed primarily as an indicator of low heterogeneity [21].

### 2.6. Data Analysis

Data analysis was performed by using the Comprehensive Meta-Analysis software version 3 for Windows. The main results were presented as standardized mean difference (SMD), relativizing both the data in relative (mL·kg^−1^·min^−1^) or absolute (L·min^−1^) $\dot{V}$O_2_max/peak units according to their standard deviation. The SMD and raw mean differences (RMD) were based on the difference between the HIIT and the SIT intervention variation (post-training minus pre-training). In addition, a subgroup analysis of RMD was performed, including only studies providing relative $\dot{V}$O_2_max/peak data (mL·kg^−1^·min^−1^). The data were summarized by using the forest plot graph, representing the results of the SMD in the overall analysis and RMD for the subgroup analysis, and a 95% confidence interval was used for both (95% CI). Moderate correlation (r = 0.5) between pre- and post-training data and a fixed effect model were adopted for the analysis.

Subgroup analyses were performed to test differences between sexes (male vs. female, excluding studies with mixed samples), age and physical-activity levels (moderately trained vs. sedentary). Furthermore, a sensitivity analysis was performed for a subgroup of high-quality studies, in accordance with TESTEX (≥10).

## 3. Results

After we removed the replicated studies, those that did not include interval-training protocols and those involving animal models, 593 studies were selected for the writing of their abstract and title. After, in a second step, studies were selected based on inclusion and exclusion criteria. Studies that presented just one of the protocols and results in graphical mode that did not allow for data extraction by software were excluded [22,23,24,25]. Finally, 19 studies were meta-analyzed (Figure 1).

The characteristics of the studies included are detailed in Table 2.

### SIT vs. HIIT

The meta-analysis showed no significant SMD between SIT and HIIT training (SMD = 0.150; 95% CI = −0.038 to 0.338; *p* = 0.119) (Figure 2). There was also no significant difference between HIIT and SIT for the subgroup of studies presenting relative $\dot{V}$O_2_max/peak data (RMD = 0.921 mL·kg^−1^·min^−1^; 95% CI = −0.185 to 2.028; *p* = 0.103, Figure 3).

Since the analyses were significant homogeneous and considerably consistent across studies for both the outcomes measured (*p* = 0.999 for SMD and *p* = 1.000 for RMD and I^2^ = 0 for both), fixed effects were assumed for both analyses. There was no significant bias of publication (*p* = 0.828), so there was no asymmetry in the funnel plot (Figure 4).

Table 3 shows that there was no confounding factor in the analysis, and the same absence of difference between HIIT and SIT was seen for different sexes, intervention durations and physical activity levels. Moreover, the sensitivity analysis for the subgroup of five studies that achieved a TESTEX score equal to or above 10 showed the same results as the overall analysis (Table 3).

## 4. Discussion

No significant differences were found for the $\dot{V}$O_2_max/peak gains when comparing HIIT and SIT protocols. There was no significant heterogeneity between the studies (*p* > 0.05), and there was no significant risk of publication bias (analysis of the funnel plot (Figure 4) and Egger’s test (*p* = 0.828)). For these reasons, the results obtained were very consistent and suggest no need for further exploration of differences between the studies. The high consistency may be explained by the proximity of the experimental design of each study, as well as by the similarity of the tests used in each study to measure the $\dot{V}$O_2_max/peak [26].

Nonetheless, we explored the potential effect of some confounding factors by subgroup and sensitivity analysis. It reinforced that different sexes, level of physical activity and duration of intervention did not influence the main effects observed on $\dot{V}$O_2_max/peak. Furthermore, there was no difference between HIIT and SIT within the subgroup of high-quality studies (TESTEX ≥ 10).

The absence of significant differences for $\dot{V}$O_2_max/peak gains when comparing HIIT and SIT corroborates the findings of Rosenblat et al. (2020) [1]. Their meta-analysis included six studies and found no differences between short, medium and long HIIT protocols for the $\dot{V}$O_2_max. In the present study, only two categories were created for the interval training group (HIIT and SIT), according to the delimitation proposed by Bucchet and Laursen (2013) [10]. For them, efforts above 60 s should be considered as HIIT, and below this time, they should be considered SIT. The cutoff used here and the higher number of studies included (19) led to greater statistical power in our analysis. Despite the differences in experimental design between the previous meta-analysis and the present one, the lack of difference between protocols was a consensus. Likely, the manipulation of the remaining training variables contributed to the similar total workload between the HIIT and SIT.

Similar gains in HIIT and SIT on the $\dot{V}$O_2_max/peak may be explained by the fact that both current protocols are predominantly focused on oxidative metabolism, since they are constituted by successive repetitions of effort/short pause that, when combined, prolong the total duration of the oxidative predominance in an exercise session [27].

Although the aim of this study was not investigate potential physiological mechanism underlying the HIIT and SIT protocols, based on current evidence, we highlighted some mechanisms related to the obtained results. Both types of protocols lead to high physiological stress, the high recruitment pattern of type I and II fibers and the vigorous muscle contraction during physical activity, which unbalances the ATP/ADP relationship and increases the activation of the PGC1-α [13,15,28]. Additionally, there is a demonstrated increase in mRNA of PGC-1 α after a SIT session with cyclists. In a biopsy of the vastus lateralis, after a 16-week HIIT intervention, there was a 138% increase of PGC1-α and an increased $\dot{V}$O_2_max that were related [29].

Although a similar increase in $\dot{V}$O_2_max/peak was found for HIIT and SIT, these protocols may lead to different magnitudes of other adaptations related to aerobic fitness, such as neuromuscular, musculoskeletal tension, cardiovascular work, anaerobic glycolytic energy and cardiac autonomic stress, causing adaptations in other variables besides $\dot{V}$O_2_max [17,30,31]. Unfortunately, for these other outcomes, there are not enough studies to be meta-analyzed. It is noteworthy that Esfarjani et al. (2007) [32] found a significant increase in the speed of the test performed on 3000 m just for the training composed by longer intervals (HIIT), as compared to a control group. Astorino et al. (2016) [33] reported an increase in cardiac output after 10 and 20 weeks of SIT, while, for HIIT, there was a stagnation in adaptation after 10 weeks. Matsuo et al. (2013) [34] demonstrated a greater SMD for $\dot{V}$O_2_max and larger effect size for stroke volume for HIIT. On the other hand, only the SIT program led to a significant increase for hematocrit.

In humans and animal models, HIIT and SIT also improve body composition and the lipid profile by decreasing LDL cholesterol (low-density lipoprotein) and increasing HDL cholesterol (high-density lipoprotein) [13,27,35,36]. The LDL protein is inversely related to nitric oxide, which is important for vessel dilation and cardiovascular conditioning. LDL decrease has been shown at 4-min HIIT intervals with concomitant increase in nitric oxide [37]. Additionally, the highest intensities achieved in interval protocols (HIIT and SIT) have been shown to cause other adaptations beyond o $\dot{V}$O_2_max in organic systems of the body, hormonal tissue and cells [12,13]. Khalafi and Symonds (2020) [38] demonstrated a decrease in body fat mass. Protocols such as Wingate have shown an increase in epinephrine and norepinephrine levels after an acute session [37], and this increase may lead to an increase in lipolytic activity [6]. In longer protocols (HIIT), a decrease in triglyceride transport has been found that may lead to a decrease in fat deposition in adipocytes [39]. A meta-analysis carried out by Maillard, Pereira and Boisseau (2017) [40] found that HIIT protocols with intensities around 90% of HR peak are effective in reducing body visceral fat. In another meta-analysis, Keating et al. (2017) [41] demonstrated that, in both HIIT and SIT, there was a decrease in body fat. In addition, studies have shown beneficial changes in the body mass, percentage of body fat mass [39], total cholesterol [35,38] and inflammatory markers [38], as well as improved insulin sensitivity, fasting insulin, adiponectin levels and endothelial function after HIIT programs [13,37,42]. Moreover, interval protocols are expected to promoted cardiovascular autonomic and functional adaptations, leading to a decrease in rest HR, improvement in premature ventricular contraction [43], reduction of systolic and diastolic and mean arterial pressure [30,44,45].

A limitation found in the present meta-analysis is the format chosen individually by each study included in the meta-analysis to report the methods used. For some of these studies, it was not possible to calculate the amount of external workload performed and not even the caloric expenditure. Thus, it was not possible to infer the impact of the differences between the volumes of the protocols in the answers found for the $\dot{V}$O_2_max/peak in the protocol comparisons. However, given the similar improvements between HIIT and SIT, such a limitation has little relevance to the adaptations of $\dot{V}$O_2_max/peak.

## 5. Conclusions

HIIT and SIT are time-efficient protocols that lead to similar gains in cardiorespiratory fitness. Thus, the choice between these training protocols should be made according to the availability of time, aptitude to perform intense physical activity and specificity of the physical conditions of each individual to practice exercise.

Future studies are encouraged to compare the effect of manipulation of other training variables, such as recovery time, number of bouts and different types of HIIT exercises (running, cycling, rowing, boxing, swimming, etc.), using equalized caloric expenditure and/or the total work performed for a comprehensive comparison between HIIT and SIT protocols.

## Figures and Tables

**Figure 1 ijerph-18-13120-f001:**
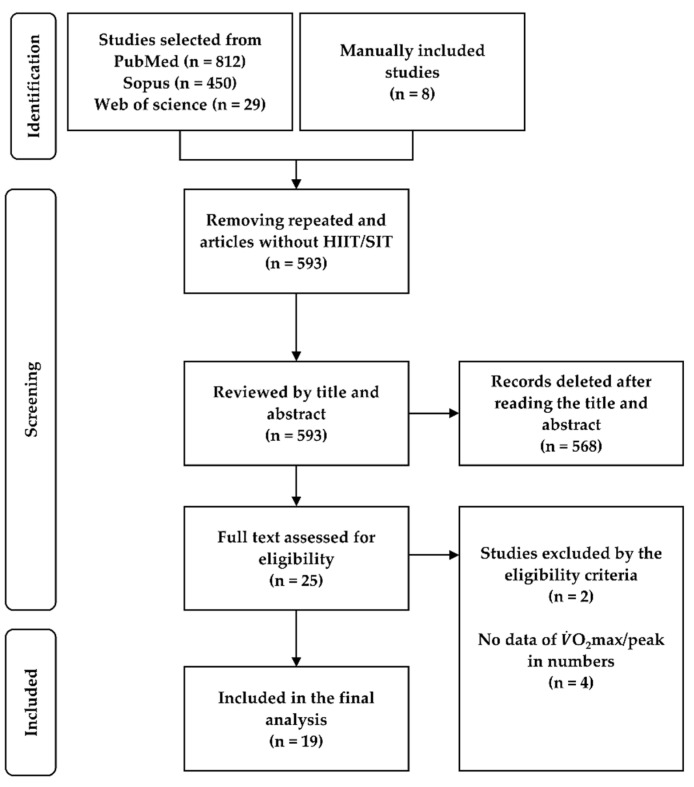
Flowchart of the studies included.

**Figure 2 ijerph-18-13120-f002:**
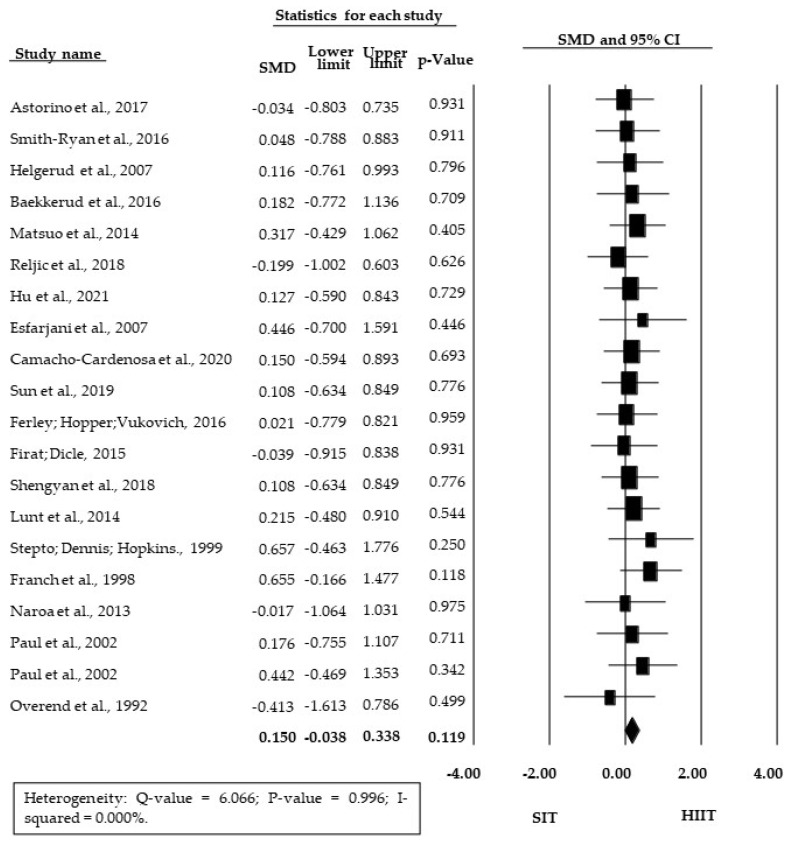
Forest plot of standardized mean difference (SMD) between HIIT and SIT; 95% CI, 95% confidence interval; SIT, sprint interval training; HIIT, high-intensity interval training.

**Figure 3 ijerph-18-13120-f003:**
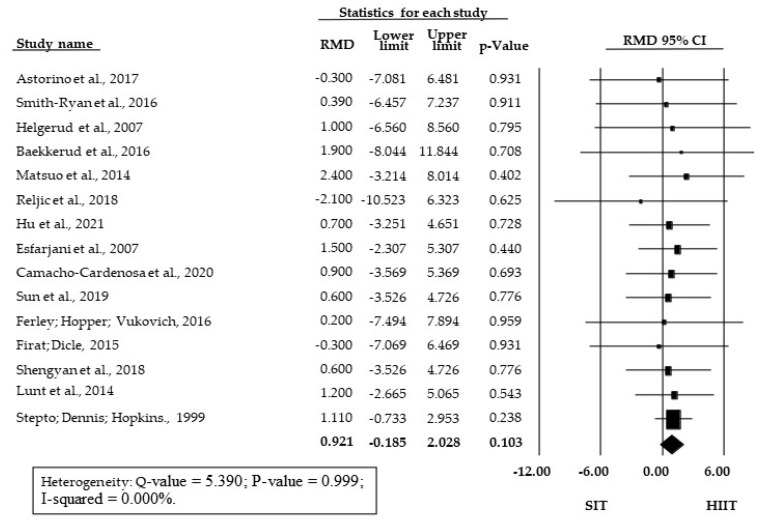
Forest Plot of raw mean differences (RMD in mL·kg^−1^·min^−1^) between HIIT and SIT; 95% CI, 95% confidence interval; SIT, sprint interval training; HIIT, high-intensity interval training.

**Figure 4 ijerph-18-13120-f004:**
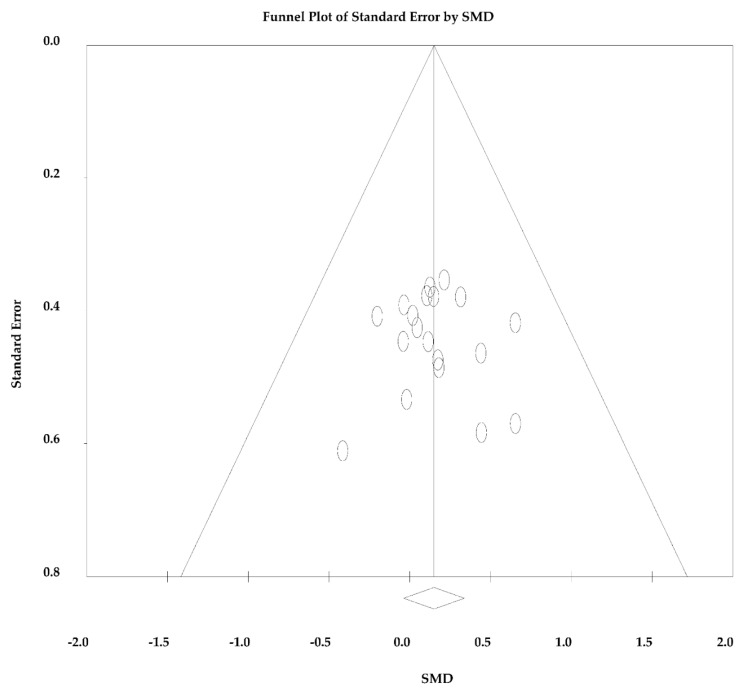
Funnel plot standardized mean difference (SMD) in mean vs. standard error of $\dot{V}$O_2_max/peak.

**Table 1 ijerph-18-13120-t001:** TESTEX Scale.

Study	Eligibility Criteria Specified	Randomization Specified	Allocation Concealment	Groups Similar at Baseline	Blinding of Assessor	Assessed in 85% of Patients	Intention-to-Treat Analysis	Between-Group Statistical Comparisons Reported	Variability Reported Outcome Measures	Activity Monitoring in Control Group	Relative Exercise Intensity Remained Constant	Exercise Volume and Energy Expenditure	Score
Astorino et al., 2017	1	0	0	1	0	2	0	2	1	1	1	0	9
Smith-Ryan et al., 2016	1	1	0	1	0	3	0	2	1	1	1	0	11
Helgerud et al., 2007	1	1	0	1	0	2	0	2	1	0	1	0	9
Bækkerud et al., 2016	1	1	0	1	0	2	0	2	1	0	1	0	9
Matsuo et al., 2014	1	1	0	1	1	1	0	2	1	1	1	1	11
Reljic et al., 2018	1	1	0	1	0	2	0	2	1	0	1	0	9
Franch et al., 1998	1	0	0	1	0	1	0	2	1	1	1	0	8
Hu et al., 2012	1	1	0	0	0	3	0	2	1	0	1	0	9
Esfarjani et al., 2007	1	0	0	1	0	0	0	2	1	1	1	0	7
Camacho-Cardenosa et al., 2020	1	1	1	1	1	0	0	0	1	0	0	1	7
Sun et al., 2019	1	1	0	1	0	1	0	2	1	0	1	0	8
Ferley, Hopper and Vukovich, 2016	1	1	0	1	0	2	0	2	1	0	1	0	9
Naroa et al., 2013	1	1	0	1	0	3	0	2	1	0	1	0	10
Overend et al., 1992	1	1	0	1	0	2	0	2	1	0	1	0	9
Firat and Dicle, 2015	1	1	0	1	0	0	0	2	1	0	1	0	7
Shengyan et al., 2018	1	1	0	1	0	1	0	2	1	0	1	0	8
Lunt et al., 2014	1	1	0	1	0	3	0	2	1	0	1	0	10
Muñoz et al., 2015	1	0	0	1	0	3	0	2	1	0	1	0	9
Paul et al., 2002	1	0	0	1	0	3	0	2	1	1	1	0	10

**Table 2 ijerph-18-13120-t002:** Study characteristics.

Study	Protocol	Population	Age (years)	BW (kg)	Height (m)	Baseline	Duration (Weeks)	No. of Sessions	Exercise Intensity	No of Reps (Start/End)	Reps Duration	Work/Rest Ratio	Δ $\dot{V}$O_2_max%	Outcomes
Smith-Ryan et al., 2016	HIIT	32-♀ sedentary	33 ± 12	88.1 ± 15.9	1.66 ± 0.53	24 ± 7	3	9	$80–100\%\;\dot{V}$O_2_max	5	120 s	2	8.83	O_2_ Peak was encountered in both groups.
	SIT								90% Power output	10	60 s	1	6.45	
Helgerud et al., 2007	HIIT	40-♀ engaged in endurance training 3x/week	25 ± 4	82 ± 12	1.82 ± 0.6	55.5 ± 7.4	8	24	90–95% HRmax	4	4 min	1.3	8.96	Both groups ↑ absolute
	SIT					60.5 ± 5.4			90–95%Hrmax	15	15 s	1	7.98	
Ferley; Hopper; Vukovich, 2016	HIIT	24: 16-♀ and 8-♂; running experience	28 ± 7	68.4 ± 8.8	1.72 ± 0.54	50.2 ± 7.2	6	18	68% Vmax	4–6	60% Tmax	NR	4.58	$\mathrm{Both}\;\mathrm{groups}\;↑\;\dot{V}$O_2_max. There was a small effect size for SIT > HIIT.
	SIT		26 ± 5	73.2 ± 12.5	1.74 ± 0.84	50.2 ± 6.9			Vmax	10–14	30 s	NR	4.98	
Astorino et al., 2017	HIIT	71: 34-♂ and 37-♀ active healthy	22 ± 5.4	69.6 ± 11.4	1.74 ± 10	39.6 ± 5.6	3–4	10	70–110% PPO	5–10	150 s	1.3	9	$\mathrm{Both}\;\mathrm{groups}\;↑\;\mathrm{relative}\;\mathrm{and}\;\mathrm{absolute}\;\dot{V}$O_2_max.
	SIT			68.5 ± 10.3	1.72 ± 8				All out	8–12	60 s	0.2	7.7	
Esfarjani, Laursen, 2007	HIIT	17-♂ moderately trained runners	23 ± 5	69.6 ± 11.4	NR	39.6 ± 5.6	10	20	60% Tmax	8	60% Tmax	1	9.16	$\mathrm{Both}\;\mathrm{groups}\;↑\;\dot{V}$O_2_max.
	SIT			68.5 ± 10.3					30 s	12	30 s	0.11	6.19	
Matsuo et al., 2014	HIIT	42-♂ sedentary	26.5 ± 6.2	63 ± 7	1.72 ± 5	41.9 ± 5.6	8	40	3 min	3	3 min	1.5	21.96	$\mathrm{Both}\;\mathrm{groups}\;↑\;\dot{V}$O_2_max. There was a larger effect size to HIIT > SIT.
	SIT			62.4 ± 5.4	171 ± 5	43.9 ± 6.7			30 s	7	30 s	1.5	15.72	
Overend; Cunningham, 1992	HIIT	17-♂ active young	25 ± 3	75 ± 9	1.77 ± 7	3.49 ± 0.26	10	40	$100\%\;\dot{V}$O_2_max	NR	3 min	1.5	9.46	$\mathrm{Both}\;\mathrm{groups}\;↑\;\dot{V}$O_2_max.
	SIT					3.15 ± 0.22			$120\%\;\dot{V}$O_2_max	NR	30 s	1	16.51	
Firat; Dicle, 2015	HIT	20-♂ national level lightweight collegiate rowers	21 ± 2	67 ± 3	1.78 ± 6	56.6 ± 5.7	4	8	90%PPO	8	2.5 min	0.833	4.91	$\mathrm{There}\;\mathrm{was}\;↑\;\dot{V}$O_2_max for all groups.
	SIT								150% PPO	10	30 s	0.11	5.53	
Franch et al., 1998	HIIT	36-♀ running experience	30.4 ± 4.8	NR	NR	54.8 ± 3.0	6	NR	$100\%\;\dot{V}$O_2_max	3–6	4 min	2	6	$\mathrm{Both}\;\mathrm{groups}\;↑\;\dot{V}$O_2_max.
	SIT								$120\%\;\dot{V}$O_2_max	30–40	15 s	1	3.6	
Sun et al., 2019	HIIT	42-♀ overweight but healthy	22 ± 2	69 ± 6	1.63 ± 5	31.5 ± 2.2	12	36	90%PPO	8–10	4 min	1.33	26.67	$\mathrm{There}\;\mathrm{was}\;↑\;\dot{V}$O_2_max for all groups.
	SIT		68 ± 7	68 ± 7	1.62 ± 3.9	31.1 ± 3.6			150% PPO	80	6 s	0.67	25.08	
Camacho-Cardenosa et al., 2019	HIIT	36-♀ running experience	30.4 ± 4.8	NR	NR	25.50 ± 4.93	12	57	90% Wmax	3–6	3 min	1	−0.47	$\mathrm{Both}\;\mathrm{groups}\;↑\;\dot{V}$O_2_max.
	SIT					25.33 ± 4.62			all-out	3–6	30 s	0.16	2.88	
Shengyan et al., 2019	HIIT	48-♀ overweight female	21.5 ± 1.8	21.5 ± 1.8	NR	31.5 ± 2.2	12	36	$90\%\;\dot{V}$o_2_peak	80	4 min	0.66	8.4	$\mathrm{Both}\;\mathrm{groups}\;↑\;\dot{V}$O_2_max.
	SIT			21.4 ± 1.1		31.1 ± 3.6			100 rpm with 1.5 kg	NR	6 s	1.33	7.8	
Reljic; Wittmann; Fischer, 2018	HIIT	34: 23-♀ and 11-♂ sedentary	30 ± 7.1	71 ± 14.2	1.67 ± 0.11	29.3 ± 7.7	8	16	85–95% HRmax	2	4 min	2	16.5	$\mathrm{Both}\;\mathrm{groups}\;↑\;\dot{V}$O_2_max.
	SIT			75.6 ± 15.4	1.73 ± 0.10				85–95% HRmax	5	1 min	1	24.14	
Hu et al., 2021	HIIT	66-♀ and ♂ sedentary	21.2 ± 1.4	26 ± 3	NR	31.9 ± 6.9	12	36	$90\%\;\dot{V}$O_2_peak	NR	4 min	1.3	20.5	$\dot{V}$O_2_max ↑ HIIT and SIT
	SIT					34.7 ± 8.7			1 kg 100 rpm	10	6 s	0.6	21.5	
Etxebarria et al., 2014	HIIT	14-♂ moderately trained	33 ± 8	78 ± 10	1.82 ± 8	58.7 ± 8.1	3	6	$80\%\;\dot{V}$O_2_peak					$\mathrm{Both}\;\mathrm{groups}\;↑\;\mathrm{small}\;\dot{V}$O_2_peak.
	SIT								Near maximal					
Baekkerud et al., 2016	HIIT	30: 18-♀ and 13-♂ sedentary	41 ± 9	91 ± 14	1.73 ± 0,08	31.9 ± 6.9	6	18	85–95% HRmax					$\dot{V}$O_2_max was ↑ HIIT then in SIT.
	SIT					34.7 ± 8.7			90% HRmax					
Lunt et al., 2014	HIIT	49: 36♀ and 13-♂ sedentary	48 ± 6	NR	NR	24.2 ± 4.8	12	36	85–95% HRmax	4	4 min	1.3	5.79	$\mathrm{There}\;\mathrm{was}\;↑\;\dot{V}$O_2_max for HIIT Walk group, but not for SIT vs. Walk.
	SIT		50 ± 8;	NR	NR	25.0 ± 2.8			All out	3	30 s	0.125	0.8	
Paul et al., 2002	HIIT	41-♀ high trained athletes	25 ± 6	75 ± 7	1.80 ± 5	64.5 ± 5.2	4	8	Pmax	8	144 s	0.5	5.20 (G1); 7.98 (G2)	$\mathrm{Both}\;\mathrm{groups}\;↑\;\dot{V}$O_2_max. There was significantly HIIT2 > SIT.
	SIT								175% PPO	12	30 s	0.11	3.05	
Overend; Cunningham, 1992	HIIT	17-♂ active young	25 ± 3	75 ± 9	1.77 ± 7	3.49 ± 0.26	10	40	$100\%\;\dot{V}$O_2_max	NR	3 min	1.5	9.46	$\mathrm{Both}\;\mathrm{groups}\;↑\;\dot{V}$O_2_max.
	SIT					3.15 ± 0.22			$120\%\;\dot{V}$O_2_max	NR	30 s	1	16.51	
Firat; Dicle, 2015	HIIT	20-♂ national level lightweight collegiate rowers	21 ± 2	67 ± 3	1.78 ± 6	56.6 ± 5.7	4	8	90% PPO	8	2.5 min	0.833	4.91	$\mathrm{There}\;\mathrm{was}\;↑\;\dot{V}$O_2_max for all groups.
	SIT								150% PPO	10	30 s	0.11	5.53	
Fahimeh, 2007	HIIT	17-♂ moderately trained runners	23 ± 5	69.6 ± 11.4	NR	39.6 ± 5.6	10	20	$v\dot{V}$O_2_max	8	60% Tmax	1	9.16	$\mathrm{Both}\;\mathrm{groups}\;↑\;\dot{V}$O_2_max
	SIT			68.5 ± 10.3					$130\%v\dot{V}$O_2_max	12	30 s	0.11	6.19	

Legend: BMI, body mass index; min, minutes; HIIT, high-intensity interval training; SIT, sprint interval training; CON, control; NR, no reported; HR, heart rate; Max, maximal; Vmax, velocity maximum; Pmax, power maximum; PPO, peak power output; v$\dot{V}$
O_2_max, velocity at the maximal oxygen uptake; $\dot{V}$O_2_max, maximal oxygen uptake; RPM, rotations per minute; MAP, maximal aerobic power; VeT, ventilation threshold, CAT, control aerobic training; ♀, women; ♂, men; PAL, physical-activity level; BW, body weight;↑, increase; ↔, no change.

**Table 3 ijerph-18-13120-t003:** Subgroup analysis.

Sex	k	SMD	LL	UL	*p*-Value	*p*-Diff
Male	7	0.099	−0.25	0.449	0.578	0.175
Female	7	0.227	−0.086	0.54	0.155	
TESTEX score 10						
≥10	4	0.172	−0.229	0.573	0.401	0.401
PAL						
Active	12	−0.14	−0.12	0.4	0.29	0.153
Sedentary	8	0.108	−0.153	0.391	0.39	
Duration (weeks)						
≥7	7	0.129	−0.174	0.431	0.404	0.175
<7	13	0.131	−0.109	0.371	0.284	

Legend: k, number of study groups; SMD, standard difference in means (mL·kg·min and L·min); *p*-value, *p*-value for significance (*p* < 0.05) between subgroups; *p*-diff, *p*-value for significance (*p* < 0.05) between categories of subgroups; LL, low limit of 95% confidence interval; UL, upper limit of 95% confidence interval; PAL, physical-activity level.

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
