# Peer review of "HIIT vs. SIT: What Is the Better to Improve V˙O2max? A Systematic Review and Meta-Analysis"

_ijerph, 2021, doi:10.3390/ijerph182413120_

Round 1

Reviewer 1 Report

HIIT vs SIT: What is the Better to Improve VO2max?— 2 A Systematic Review and Meta-Analysis

Abstract

Occasional grammatical errors e.g. ‘time- eficiente strategy’. Use of ‘the’ inappropriately e.g. ‘the High-Intensity Interval Training (HIIT)’

Intro – needs to be more specific in the language used e.g. Line 47 – what is meant by maximum? Intensity equivalent to Vo2max/peak or maximum power?

Line 60 – I’m not sure that the authors have really established in the introduction that the hypothesis is valid. The introduction is too vague – needs greater focus.

Materials and Methods

I would expect a greater number of databases to have been searched, rather than a single one and also would expected details of attempts to interrogate the ‘grey literature’.

Lacks details of the participant inclusion criteria – rather vague. The Pedro scale is not the most appropriate quality screening tool for the training interventions detailed in the review - the TESTEX was developed specifically for exercise interventions and is more suitable.

Results

I would expect some summary tables in a systematic review and meta-analysis.

There appears little sensitivity analysis of the meta-analysis findings e.g. did study quality (as assessed by the Pedro scale) influence the findings? Did sample size? Training duration? Participant’s initial fitness etc? No real subgroup analysis.

Discussion

The discussion explores the potential physiological mechanisms that might have been expected to result in differing training responses. However, I do not see how this systematic review adds much to the literature.

The systematic review and meta-analysis seems opportunistic rather than clearly designed and planned. For example, I would expect it to have been registered in Prospero before data collection, thus keeping the findings transparent and reducing the risk of outcome and reporting bias. The literature search was only performed 4 weeks ago and in my opinion, this brief time does not allow sufficient time to perform a rigorous systematic review and meta-analysis as evidenced by the review submitted.

Author Response

Abstract

Occasional grammatical errors e.g. ‘time- eficiente strategy’. Use of ‘the’ inappropriately e.g. ‘the High-Intensity Interval Training (HIIT)’

Answer: Some typing erros were  corrected in the new document.

Abstract

Line 15:“In this sense, interval training was suggested as a time efficient...”

Introduction

Line 46:“In this sense, High Intensity Interval Training (HIIT)[…].”

Line 47:“HIIT and SIT training are time efficient[...].”

7  Intro – needs to be more specific in the language used e.g. Line 47 – what is meant by maximum? Intensity equivalent to Vo2max/peak or maximum power?

Answer: Line 47: This definition was considered to include several types of HIIT prescription (HIIT and SIT) which can be based on different measurenment relative to individual maximal capacity (Buchet and Laursen, 2013). Most of the selected studies in our meta-analysis were based on this definition.

Line 60: “[…]efforts lasting 30 seconds at all-out intensity, to protocols using intensities close to maximum (according with the diferents forms of prescription e.g. VO2max, maximum power output, Vmax etc.)[10][…]”

Line 60 – I’m not sure that the authors have really established in the introduction that the hypothesis is valid. The introduction is too vague – needs greater focus.

Answer: The hypothesis of this study is that the different duration in HIIT effort phase lead to different metabolic and organic system demands, thus leading to different adaptations in relation to O2max.

Line 76: “The gains in the O2max have been associated with intensities close to that of the O2max [10-11,16]. This phenomenon is dependent on the increase of the exercise intensity, duration of the stimulus, the requirement (recruitment/activation) of types I and II muscle fibers and the use of energetic substrates [17]. For example, while bouts at maximum/supramaximal intensities with short duration should also affect the neuromuscular system, bouts at submaximal/near to maximum intensities mainly improve cardiovascular and cardiorespiratory fitness [10]. These findings led us to hypothesize that different duration in HIIT effort phase lead to different metabolic and organic system demands, thus leading to different adaptations in relation to O2max.”

Materials and Methods

I would expect a greater number of databases to have been searched, rather than a single one and also would expected details of attempts to interrogate the ‘grey literature’.

Answer: We have included searches in two more databases as also suggested by the other reviwers. Please, check the modification in the text.

Line 93: “Searches were conducted in Octuber 2021 using articles from PubMed, Scopus and Web of Science databases[...]”

Lacks details of the participant inclusion criteria – rather vague.

Answer: More details have been added.

Line 119: No restrictions were applied to the different ages, sex, body composition, comorbidities and levels of physical activity of the subjects included in each survey”.

The Pedro scale is not the most appropriate quality screening tool for the training interventions detailed in the review - the TESTEX was developed specifically for exercise interventions and is more suitable.

Answer: We thank the reviwer for this suggestion. The Pedro scale has been replaced by TESTEX scale in the new document according to the suggestion. Please, check Line 106.

Results

I would expect some summary tables in a systematic review and meta-analysis.

Answer: Line 155: We have included the Table 2 with the studies characterization.

Discussion

The discussion explores the potential physiological mechanisms that might have been expected to result in differing training responses. However, I do not see how this systematic review adds much to the literature.

Answer: In fact, our objective were not investigate potential physiological mechanism underling the HIIT and SIT protocols. However, based on current evidences, we highlight some mechanisms related to the obtained results. More studies and future meta-analyses are needed to better understand the physiological mechanisms related to the similar adaptations in cardiorespiratory fitness when comparing HIIT vs SIT.

The systematic review and meta-analysis seems opportunistic rather than clearly designed and planned. For example, I would expect it to have been registered in Prospero before data collection, thus keeping the findings transparent and reducing the risk of outcome and reporting bias. The literature search was only performed 4 weeks ago and in my opinion, this brief time does not allow sufficient time to perform a rigorous systematic review and meta-analysis as evidenced by the review submitted.

Answer:  We understand the reviewer's concern, but we emphasize that this is not an opportunistic sistematic review and metanalysis and it was conducted with a concerted effort by the authors of this study. This review has been going on for a long time and has been updated a few times before the final submission, just as we have just done including searches in new databases as suggested by the reviewers. Throughout these updates the main results have not changed, requiring few textual modifications.

Unfortunately, we just don't have our sistematic review and metanalysis registered in Prospera. However we ensured it was rigorously conducted following the guidelines of the Preferred Reporting Items for Systematic and Meta-Analyses (PRISMA).

Reviewer 2 Report

A systematic review: " HIIT vs SIT: What is the better to improve VO2max? A systematic review and meta-analysis” was submitted for review. From a methodological point of view, the work is very interesting, taking into account the most popular training improving cardiorespiratory fitness. This type of work is typically needed recently as I have not found a similar meta-analysis. The introduction to the work is written correctly, but for the reader who would like to expand their knowledge and do not work in sports, the data on the characteristics of training are too general and should be expanded.

The methodology of the research is very reliably and correctly presented. However, in my opinion, Figure 1 should be included in the methodology part, not the results. I also believe that table 2 in supplementary materials should be reduced and introduced into the main part of the work.

The dissertation discussion is written correctly. What also draws attention is the presentation of the limitations of the work, which shows the maturity and reliability of the authors. The conclusions are more of a summary of the work and should be redrafted.

I also pay attention to references: 7, 8, 11, 17 and 30 - asking the authors to quote newer literature.

Author Response

Reviewer #2:

A systematic review: " HIIT vs SIT: What is the better to improve VO2max? A systematic review and meta-analysis” was submitted for review. From a methodological point of view, the work is very interesting, taking into account the most popular training improving cardiorespiratory fitness. This type of work is typically needed recently as I have not found a similar meta-analysis. The introduction to the work is written correctly, but for the reader who would like to expand their knowledge and do not work in sports, the data on the characteristics of training are too general and should be expanded.

Answer: Please, check the changes in the text.

Line 49: “HIIT consists of performing repeated series of high intensity efforts (usually between the second ventilatory threshold and the O2max/peak) interspersed with periods of low intensity recovery/or pause [1,7,9]. In this perspective, another protocols has been emerged for variations of HIIT using intensities and/or loads above the O2max/peak, for very short periods of time in high intensity. Here this short bout protocols was named Sprint Interval Training (SIT) [1,10].”

The methodology of the research is very reliably and correctly presented. However, in my opinion, Figure 1 should be included in the methodology part, not the results. I also believe that table 2 in supplementary materials should be reduced and introduced into the main part of the work.

Answer: Thank you for the comment. We have inserted the figures and tables acording to PRISMA check list for reporting the meta-analysis results (Moher; Liberati; Tetzlaff; Altman, 2009). Supplementary Table 2 was inserted as a regular table in the manuscript.   

The dissertation discussion is written correctly. What also draws attention is the presentation of the limitations of the work, which shows the maturity and reliability of the authors. The conclusions are more of a summary of the work and should be redrafted

Answer: Thank you for the comment.  We rewrite the conclusion.

Line 427:  “In this review was verified that both HIIT and SIT are time efficient protocols for improving similarly cardiorespiratory fitness. Thus, the choice between these training protocols should be made according to the availability of time, aptitude to perform intense physical activity and specificity of the physical conditions of each individual to practice exercise.

Future studies should be encouraged to investigate the effect of HIIT and SIT from the perspective of other training variables, such as recovery time, number of fights and different types of HIIT exercises (running, cycling, rowing, boxe, swiming, etc.).”

Reviewer 3 Report

This systematic review and meta-analysis describe two training methods, HIIT and SIT, and their effects on VO2max. The discussion lists the adaptations achieved with high-intensity exercise that increase VO2max. It is summarized that both exercise protocols cause similar increases in VO2max and are recommended for training.  

pp 39-41 In this perspective, another nomenclature has been used for variations of interval training protocols using intensities and/or loads above the ?̇O2max/peak, for very short periods of time such as Sprint Interval Training (SIT) [1,10].

here, it needs a piece of information about recovery between bouts, eg. kind, and ratio, as mentioned in the HIIT

pp 69-70 The descriptors for duration of effort were: exercise dose; short extent; long extent; long time; short time; long volume; short volume; long volume

pp 93-105 2.2.3 Protocols .....

in my opinion, it is useful to mention the type of recovery and E/R ratio in each of the exercise categories as mentioned for the duration

p 177 becase of remaning / because of remaining

Author Response

Reviewer #3:

This systematic review and meta-analysis describe two training methods, HIIT and SIT, and their effects on VO2max. The discussion lists the adaptations achieved with high-intensity exercise that increase VO2max. It is summarized that both exercise protocols cause similar increases in VO2max and are recommended for training.  

pp 39-41 In this perspective, another nomenclature has been used for variations of interval training protocols using intensities and/or loads above the ?̇O2max/peak, for very short periods of time such as Sprint Interval Training (SIT) [1,10].

Answer: pp39-41 Please, check the modification in the text.

Line 54:“[…]periods of time as Sprint Interval Training (SIT) [1,10].”

here, it needs a piece of information about recovery between bouts, eg. kind, and ratio, as mentioned in the HIIT

Answer: This information have been reported in Table 2 of studies characterization.

Line 156: Table 2. Studies characteristics

pp 69-70 The descriptors for duration of effort were: exercise dose; short extent; long extent; long time; short time; long volume; short volume; long volume

Answer: Please, check the modification in the text.

Line 100: “The descriptors for duration of effort were: exercise dose; short extent; long extent; long time; short time; long volume; short volume”

pp 93-105 2.2.3 Protocols .....

in my opinion, it is useful to mention the type of recovery and E/R ratio in each of the exercise categories as mentioned for the duration

Answer: We thank the reviewer for this observation. This information can be verified in Table 1. Please, check the modifications in the text.

Line 156: Table 2. Studies characteristics”

p 177 becase of remaning / because of remaining

Answer: P 177 Please, check the modification in the text.

Line 347“[...]because of remaining[...]”

Reviewer 4 Report

The aim of this systematic review and meta-analysis is to learn which interval training protocol, HIIT or SIT, promotes a greater gain in cardiorespiratory fitness (?O2max/peak).

Although it is an interesting paper, some modifications should be made to the text.

Here are some contributions:

Add "systematic" to the title. “Systematic review”.

Line 47, the intensity is not maximal, but supramaximal, because these are intensities higher than VO2 max.

Brief introduction. It would be appropriate to add a paragraph referring (without going into the results obtained) to some of the available papers on the case, and to demonstrate the cardiorespiratory improvement with the two methods.

It is necessary to include more databases (SCOPUS, WOS...) in the searching.

Which were the search phrases used in the study? it is necessary to include them.

In Figure 1, specify a little more about the reasons for exclusion of the 369 papers.

All 21 articles should be included in the review and the final 18 in the meta-analysis. Even if not all the data are available for the meta-analysis, these three articles should be included in the review.

It is necessary to include a table with all the selected articles and their characteristics: population, intervention duration, variables measured…

In the results section, it would be advisable to briefly describe the results obtained.

Line 195, 209, 212…use abbreviation? Review all the test.

Congratulations for your work.

Author Response

Reviewer #4:

Line 47, the intensity is not maximal, but supramaximal, because these are intensities higher than VO2 max.

Answer: Line 61,  We agree with the reviewer. Please, check the modification in the text.

Brief introduction. It would be appropriate to add a paragraph referring (without going into the results obtained) to some of the available papers on the case, and to demonstrate the cardiorespiratory improvement with the two methods.

Answer: Line 66: When comparing this protocols, Matsuo et al., 2014 demonstrated a significant increase in O2max after 8 weeks of HIIT and SIT, evidencing that both programs are effective to improve cardiorespiratory fitness [12].”

It is necessary to include more databases (SCOPUS, WOS...) in the searching.

Answer: Line 94:  We agree with the reviewer. Please, check the modification in the text. We have included searches in these two additional database as suggested.

In Figure 1, specify a little more about the reasons for exclusion of the 369 papers.

Answer: Please check the correction in the text.

Line 194:“After removes the replicated studies, those that not included interval training protocols, and involving animal models 593 were selected for riding of abstract and title. After, in a second step studies were selected based on inclusion and exclusion criteria. Studies that presented just one of the protocols and results in graphical mode that did not allow data extraction by software.”

All 21 articles should be included in the review and the final 18 in the meta-analysis. Even if not all the data are available for the meta-analysis, these three articles should be included in the review.

Answer: The excluded estudies was included in the text and references. Please check the correction in the text.

Line 197:“[…] results in graphical mode that did not allow data extraction by software [45-48].”

It is necessary to include a table with all the selected articles and their characteristics: population, intervention duration, variables measured…

Answer: We have included a Table 2 for studies characterization. Please check the modification in the text.

Line 156: Table 2. Studies characterization”

In the results section, it would be advisable to briefly describe the results obtained.

Answer: We chose to add Table 2 (Study characteristics) in the manuscript as a regular table in order not to make the results section extensively long. please check the line 156.

Line 195, 209, 212…use abbreviation? Review all the test.

Answer: Line 195, 209,212 The abreviation was reviewed.

Round 2

Reviewer 1 Report

Review of systematic review/meta-analysis

Overall there are fewer errors/omissions than the first draft.

There are still several typos e.g. ‘trainig’, ‘simalarity’ and grammatical errors.

It seems surprising that no additional papers were found if two additional databases were searched.

The TESTEX (not TEXTES) scale could be improved – the referencing is inconsistent – why are initials there and why does the referencing format change so often e.g. Paul B. Laursen et al., 2002, Hu M, et al.,2012 – shows a lack of attention to detail.

Table 2 should be in results section and is poorly formatted and thus over 6 pages. Are height and weight details needed or would BMI capture these more concisely? At the very least the pages should change to landscape layout for these summary tables. Is 2 decimal places needed for % change? Units of VO2max are not (ml.kg.min).

Figure 3 – again a lack of attention to detail with respect to referencing. This is also present in the text in the Discussion section.

Figure 4- the word Fixed appears in the middle of the author list.

There is still no sensitivity analysis to examine if important issues such as trial duration, study quality (according to the TESTEX), baseline fitness etc impacted the results. Could the absence of differences be due to the heterogeneous (not statistically) nature of the studies in the 19 selected – ie difference in study protocols, baseline fitness etc.

One of my previous comments was:

The discussion explores the potential physiological mechanisms that might have been expected to result in differing training responses. However, I do not see how this systematic review adds much to the literature.

Answer: In fact, our objective were not investigate potential physiological mechanism underling the HIIT and SIT protocols. However, based on current evidences, we highlight some mechanisms related to the obtained results. More studies and future meta-analyses are needed to better understand the physiological mechanisms related to the similar adaptations in cardiorespiratory fitness when comparing HIIT vs SIT.

The authors seem to have misunderstood my comment, so I would like to clarify it. I was referring to the fact that the Discussion largely focuses on the physiological mechanisms which might underly the differences in outcomes. I see this as an error, since as the authors state above, it was not the objective of the article. My point was that the Discussion should focus on the outcomes of the systematic review and meta-analysis – examining sensitivity etc whereas it spends more time discussing physiological mechanisms, which are not the focus of the paper.

Overall, the authors have improved the paper, for example by adding results tables. It was surprising that the initial submission was missing standard details such as these which are expected in any systematic review meta-analysis. I am still concerned that the literature has not been thoroughly searched. The lack of attention to detail in the article concerns me e.g. the tables/figures have at least 3 different ways of referencing. The Discussion section has had little alteration and does not explore the results of the systematic review/meta-analysis in sufficient detail in my opinion.

Author Response

Referee 1

  • Overall there are fewer errors/omissions than the first draft.

There are still several typos e.g. ‘trainig’, ‘simalarity’ and grammatical errors.

Answer: We did one more revision in the whole text and also sent to English language review to solve possible persisting errors.

  • It seems surprising that no additional papers were found if two additional databases were searched.

Answer: We have some experience in systematic reviews and for some specific topics, such as HIIT and O2max we have seen that PubMed usually contains between 90 and 100% of the studies. In this case we just found 1 new study. Usually, the studies that are in other databases they will be also on PubMed. Furthermore, regarding exercise training effects, most journals that are not indexed on PubMed will be lower impact factor journals and could be not advantageous to explore some databases. In any case, we attached in this letter the full syntax of the searches.

  • The TESTEX (not TEXTES) scale could be improved – the referencing is inconsistent – why are initials there and why does the referencing format change so often e.g. Paul B. Laursen et al., 2002, Hu M, et al.,2012 – shows a lack of attention to detail.

Answer: Line 228 — We reviewed it again and we believe now it is correct.

  • Table 2 should be in results section and is poorly formatted and thus over 6 pages. Are height and weight details needed or would BMI capture these more concisely? At the very least the pages should change to landscape layout for these summary tables. Is 2 decimal places needed for % change? Units of VO2max are not(ml.kg.min).

Answer: Page 229 — We agree that the table was not in the best format and to be honest we have changed it many times before the submission of the manuscript. Now we tried a different format, that make it at least more concise and easier to read and we hope you all approve it. In any case, we accept further suggestion, and we will be more than glad to find a better option whether there is a better option. Regarding Height and Weight most of studies reported these values, however not all of them reported BMI, so that is why we opted to show their original values instead of present a rough estimation calculated from the group means.

  • Figure 3 – again a lack of attention to detail with respect to referencing. This is also present in the text in the Discussion section.

Answer: Line 382 —Thank you for this second review to fix something we should have done from the beginning; we did our best to solve it now.

  • Figure 4- the word Fixed appears in the middle of the author list.

Answer: Thank you to point it out, we corrected it.

  • There is still no sensitivity analysis to examine if important issues such as trial duration, study quality (according to the TESTEX), baseline fitness etc impacted the results. Could the absence of differences be due to the heterogeneous (not statistically) nature of the studies in the 19 selected – ie difference in study protocols, baseline fitness etc.

Answer: Line 372 — Thanks to bring it to discussion, we did subgroups analysis in the beginning of this project and we did not find any difference, and as there wasn`t statistical heterogeneity we believed it could be criticized if we presented the non-significant differences between subgroups. To avoid any doubt, we now added the subgroup analysis for duration of intervention, level of physical activity, and sex. We did not present age comparison since we just have two studies middle-aged adults (results with SMD 0.204 [LL -0.358:UL 0.765], p = 0.477), while the rest were young adults (results com SMD 0.121 [LL -0.079:UL 0.320], p = 0.235) and there was not trend for differences among them (p = 0.175). Also, we showed sensitivity analysis for the subgroup of 4 studies that got TESTEX score equal or above 10. Despite we did not saw any difference from the overall analysis, we can show here if there was any potential effect of studies quality in the main effect it should be in this cut-off that we used.

  • One of my previous comments was:

The discussion explores the potential physiological mechanisms that might have been expected to result in differing training responses. However, I do not see how this systematic review adds much to the literature.

The authors seem to have misunderstood my comment, so I would like to clarify it. I was referring to the fact that the Discussion largely focuses on the physiological mechanisms which might underly the differences in outcomes. I see this as an error, since as the authors state above, it was not the objective of the article. My point was that the Discussion should focus on the outcomes of the systematic review and meta-analysis – examining sensitivity etc whereas it spends more time discussing physiological mechanisms, which are not the focus of the paper.

Overall, the authors have improved the paper, for example by adding results tables. It was surprising that the initial submission was missing standard details such as these which are expected in any systematic review meta-analysis. I am still concerned that the literature has not been thoroughly searched. The lack of attention to detail in the article concerns me e.g. the tables/figures have at least 3 different ways of referencing. The Discussion section has had little alteration and does not explore the results of the systematic review/meta-analysis in sufficient detail in my opinion.

Answer: We agree that we discussed more about the potential physiological mechanisms them expected for a meta-analysis, and more than we usually do. We did it because there is a lot of theoretical hypotheses about what should cause the difference between those two protocols that were not confirmed with the analysis. We believe these results are very important to bring a consensus and reinforce that both protocols lead to the same benefits regarding O2max and thus is very applied for sedentary or physically active individuals who want to improve cardiorespiratory fitness. We also tried to make all the discussion more concise in regard to physiological mechanisms.  In addition, with the new subgroup and sensitivity analysis we increase the focus to the possible confounding factors that could affect the results but did not. So the consistency of our results was just reinforced by these new analysis. We hope it became more appropriate now.

Reviewer 4 Report

The text has been considerably improved.

Congratulations to the authors.

Author Response

Referee 2

  • The text has been considerably improved. Congratulations to the authors.

Answer: Thank you so much for the opportunity, we did our best to improve it one more time.
